# Effect of Sex on Clinical Outcomes in Patients with Coronavirus Disease: A Population-Based Study

**DOI:** 10.3390/jcm10010038

**Published:** 2020-12-24

**Authors:** Kyu Hyang Cho, Sang Won Kim, Jong Won Park, Jun Young Do, Seok Hui Kang

**Affiliations:** 1Division of Nephrology, Department of Internal Medicine, College of Medicine, Yeungnam University, Daegu 705-717, Korea; chokh@yu.ac.kr (K.H.C.); karismatajongwon@gmail.com (J.W.P.); jydo@med.yu.ac.kr (J.Y.D.); 2Medical Research Center, College of Medicine, Yeungnam University Medical Center, Daegu 705-717, Korea; kimsw3767@ynu.ac.kr

**Keywords:** coronavirus disease, mortality, sex, clinical outcome

## Abstract

Background: This study aimed to evaluate the association between sex and clinical outcomes in patients with coronavirus disease (COVID-19) using a population-based dataset. Methods: In this retrospective study, insurance claims data from the Korea database were used. Patients who tested positive for COVID-19 were included in the study. All diseases were defined according to the International Classification of Diseases 10th revision. During follow-up, the clinical outcomes, except mortality, were assessed using the electrical codes from the dataset. The clinical outcomes noted were: hospitalization, the use of inotropics, high flow nasal cannula, conventional oxygen therapy, mechanical ventilation, extracorporeal membrane oxygenation, development of acute kidney injury, cardiac arrest, myocardial infarction, acute heart failure, pulmonary embolism, and disseminated intravascular coagulation after the diagnosis of COVID-19. Results: A total of 7327 patients were included; of these, 2964 patients (40.5%) were men and 4363 patients (59.5%) were women. There were no significant differences in the Charlson comorbidity index score between men and women in the same age group. The incidence of mortality and clinical outcomes was higher among men than among women. The mortality rate was the highest for the populations aged 50–64 or ≥65 years. The subgroup analyses for age, diabetes mellitus, or hypertension showed favorable results for patient survival or clinical outcomes for women compared to men. Conclusion: Our population-based study showed that female patients with COVID-19 were associated with favorable outcomes. Furthermore, the impact of sex was more evident in patients aged 50–64 or ≥65 years.

## 1. Introduction

Coronavirus disease (COVID-19), caused by severe acute respiratory syndrome coronavirus 2 (SARS-CoV-2), is a global pandemic. An outbreak of COVID-19 was also witnessed in South Korea. As of 8 July 2020, 13,244 patients were confirmed to be positive for SARS-COV-2 infection in South Korea [1]. The number of deaths among the confirmed cases was 285 (2.2%). Although the number of new confirmed cases has decreased compared to that noted in March 2020, confirmed cases are being diagnosed to date. Epidemiologic studies published from countries where the outbreak was noted have reported various factors associated with the prognosis of COVID-19 patients.

The first epidemiologic study involving 41 COVID-19 patients reported lower rates for intensive care unit care for women than for men; however, the difference was not statistically significant [2]. Subsequent clinical studies with large sample sizes have reported a positive association between female sex and favorable prognosis for COVID-19 patients [3,4,5]. Docherty et al. evaluated 20,133 patients with COVID-19 in the UK and reported that the hazard ratio for female patients was 0.81 compared to male patients [5]. The Global Health 50/50 initiative, using data from >20 countries, recently reported higher mortality among men than among women [6]. Previous studies have suggested that sex-related differences in the prognosis of COVID-19 patients may be associated with the difference in the expression of angiotensin-converting enzyme-2 (ACE-2), immune responses, comorbidities, socioeconomic factors, or environmental factors, such as smoking or alcohol consumption [6,7,8]. However, statistically significant results indicating an association between underlying comorbidities and sex-based differences in the outcome of COVID-19 patients have not been reported. Previous studies have reported the association between sex and clinical outcomes; however, the most relevant results were obtained during the analyses of other prognostic factors or analyses using aggregated data for sex. A few studies have focused on the association between sex and clinical outcomes as the primary objective. However, further epidemiologic data on sex differences in the outcomes of COVID-19 patients would aid in identifying the association between sex and clinical outcomes and in determining the pathogenesis of the disease or optimal treatment strategies for patients with COVID-19. This study aimed to evaluate the association between sex and clinical outcomes in patients with COVID-19 using a population-based dataset.

## 2. Experimental Section

### 2.1. Data Source

This retrospective study was conducted utilizing the database of insurance claims from the Health Insurance Review and Assessment Service (HIRA) of Korea, which is a government affiliated organization. In South Korea, 97% of the population is obliged to enroll in the National Health Insurance Program and pay the insurance according to income, regardless of medical care [9]. Patients who receive medical care, except non-essential care such as cosmetic surgery, pay approximately 5~30% of the total costs to the hospital that performs the relevant procedures. The hospital then submits the claim data to the HIRA service, who reimburse the remaining cost (approximately 70~95% of the total cost) except the patient’s payments. The claim data include diagnosis using the International Classification of Diseases, 10th revision (ICD-10) code, procedures, prescription records, and simple demographic data. Most of the population not included in the National Health Insurance Program are included in the Medical Aid Program. Patients included in the Medical Aid Program do not pay the hospital; the hospital reimburses all costs through claims to HIRA using the same approach as that used by the National Health Insurance Program. Claim data were not originally developed for medical studies. However, all hospitals and almost the entire population in South Korea uses the HIRA system. The data in the HIRA system include numerous demographics and diagnostic codes. If claims are not submitted to the HIRA system in time, the data for medical care do not coincide with actual medical care. However, many studies have used the data for conducting population-based studies. The HIRA recently identified all patients that visited for diagnosis of COVID-19 from 1 February 2020, to 15 May 2020 They compiled the data of insurance claims during the last three years from 1 January 2017, to 15 May 2020 and are providing the same to researchers after anonymization and de-identification [10]. 

### 2.2. Study Population and Variables

Confirmed patients were included among all participants who underwent COVID-19 testing. Patients aged <18 years or those who had undergone maintenance dialysis were excluded from the study. Baseline characteristics included age, sex, time of diagnosis of COVID-19, and comorbidities. The follow-up duration, death at the time of end-point of follow-up, clinical outcomes except death (hospitalization, use of inotropes, conventional oxygen therapy, high flow nasal cannula (HFNC), mechanical ventilation (MV), extracorporeal membrane oxygenation (ECMO), development of acute kidney injury (AKI), cardiac arrest, myocardial infarction (MI), acute heart failure (AHF), cerebrovascular disease, pulmonary embolism, or disseminated intravascular coagulation) were assessed after the diagnosis of COVID-19.

The ICD-10 was used to define all diseases, including COVID-19. COVID-19 patients were defined as patients with diagnostic codes for COVID-19 (B342, B972, Z208, Z290, U18, U181, Z038, Z115, U071, or U072) during the COVID-19 epidemic. The presence of comorbidities was evaluated during the last year before the diagnosis of COVID-19 and defined as codes from Quan et al. [11,12]. Seventeen comorbidities were included in the Charlson comorbidity index (CCI): MI, congestive heart failure, peripheral vascular disease, cerebrovascular disease, dementia, chronic pulmonary disease, rheumatologic disease, peptic ulcer disease, mild liver disease, diabetes mellitus (DM) without chronic complications, hemiplegia, renal disease, DM with chronic complications, any malignancy, moderate to severe liver disease, metastatic tumor, and acquired immune deficiency syndrome. Finally, the CCI score was calculated from the abovementioned 17 comorbidities. During follow-up, clinical outcomes except death were defined using Electronic Data Interchange or ICD-10 codes from HIRA. The codes were as follows: M0040 for conventional oxygen therapy, M0046 for HFNC, M5850 or M5857~M5860 for MV, O1901~O1904 for ECMO, O7031~O7035 or O7051~O7055 for dialysis, I10, M5873~M5877, or M5880 for cardiac arrest, I21, I22, I252, M655x~M657x, OA631x~OA639x, OB631x–OB639x, OA641x, OA642x, OA647x, O0161x–O0171x, or O1641x–O1647x for MI, I110, I130, I132, I255, I420, I425, I428, I429, I43, or I50 for AHF, I26 for pulmonary embolism, and D65 for disseminated intravascular coagulation. The use of inotropes was defined as the use of norepinephrine, epinephrine, vasopressin, dopamine, or dobutamine after diagnosis of COVID-19. Patients requiring dialysis after the diagnosis of COVID-19 were considered as having AKI.

### 2.3. Statistical Analyses

Data were analyzed using SAS Enterprise Guide version 7.1 (SAS Institute, Cary, NC, USA). Categorical variables were expressed as numbers and percentages, and continuous variables as mean ± standard deviation. The Pearson χ^2^ test or Fisher’s exact test was used in analyzing categorical variables. For continuous variables, the means were compared using the Student’s *t*-test or one-way analysis of variance, followed by Bonferroni’s post-hoc comparison. The survival estimates were calculated using the Kaplan–Meier curve and Cox regression analyses, while the log–rank test was used to determine *p*-values for the comparison of survival curves. Multivariate Cox regression analyses were adjusted for age, sex, CCI score, and hypertension. Clinical outcomes in acute or chronic diseases are highly influenced by underlying comorbidities, such as DM, hypertension, cerebrovascular disease, or heart disease. Although our study focused on the impact of sex in prognosis in COVID-19 patients, adjustment for underlying comorbidities will be important for identify whether sex is independently associated with prognosis. Therefore, we used CCI score as a merged indicator. Most comorbidities associated with prognosis were included in the CCI score, but hypertension as an important disease was not included in the CCI score. Therefore, we considered hypertension as an additional covariate. Additionally, logistic regression analyses were performed to evaluate the independent variables for clinical outcomes. In our study, cases with follow-up loss were not observed. Survival analyses were performed without censored data. For all outcome-related variables except survival, the patients with one clinical outcome were prone to risk of the other clinical outcomes and each outcome was independently analyzed without termination of follow-up. *p* < 0.05 was considered to be statistically significant.

### 2.4. Ethics Statement

We conducted a retrospective study using these data. The study was approved by the Institutional Review Board of the Yeungnam University Medical Center (IRB No: YUMC 2020-04-128). The board waived the need to obtain informed consent. The study was conducted following the principles of the Declaration of Helsinki.

## 3. Results

### 3.1. Clinical Characteristics of the Participants

A total of 234,427 patients underwent laboratory testing for COVID-19. Among them, 7590 patients (3.2%) were diagnosed with COVID-19. Participants aged <18 years (*n* = 249) or those who underwent maintenance dialysis (*n* = 14) were excluded. Finally, 7327 patients were included; of these, 2964 patients (40.5%) were men and 4363 patients (59.5%) were women (Table 1). The mean age of the men and women was 45.7 ± 19.3 and 47.9 ± 18.7 years, respectively (*p* = 0.093). The prevalence of peripheral vascular disease, dementia, chronic pulmonary disease, connective tissue disease, peptic ulcer disease, or any malignancy was higher in women than in men. The prevalence of hypertension, hemiplegia, or MI was higher in men than in women. In addition, we evaluated the cause of prevalent hemiplegia and found 11 (22%) men and 10 (20.4%) women developed hemiplegia due to hemorrhagic stroke. Hemiplegia caused by ischemic stroke was seen in 26 (52%) men and 33 (67.3%) women. There was no significant difference in type of stroke in patients with prevalent hemiplegia between the sexes (*p* = 0.178). The CCI scores among the men and women were 1.3 ± 1.9 and 1.4 ± 1.9, respectively (*p* = 0.039). The CCI score was slightly higher for women than for men.

### 3.2. Differences in Clinical Outcomes According to Sex

A total of 223 (3.0%) patients dies died during follow-up. One hundred and twenty (4.0%) men and 103 (2.4%) women died during the follow-up (*p* < 0.001). Kaplan–Meier curves showed that the male sex was associated with poor survival in COVID-19 patients (*p* < 0.001; Figure 1). Univariate and multivariate Cox-regression analyses showed that men had higher mortality risk compared to women (Table 2).

The number of patients with hospitalization was 2800 (94.5%) in men and 4111 (94.2%) in women, respectively (*p* = 0.681). There were 17 (0.6%) men and 10 (0.2%) women who developed AKI (*p* = 0.017; Table 3). The rate of use of inotropes, conventional oxygen therapy, HFNC, MV, cardiac arrest, and MI were greater in men than in women. Multivariate logistic regression analyses showed that the odds ratios for the use of inotropes, conventional oxygen therapy, HFNC, MV, AKI, cardiac arrest, and MI were greater for men than for women (Table 4). The odds ratios for ECMO and AHF were greater for men than for women, but no statistical significance was observed. Among patients without prevalent cerebrovascular disease, 15 (0.5%) men and 5 (0.1%) women had newly developed cerebrovascular disease (*p* = 0.002). Regarding incident pulmonary embolism, 13 (0.4%) men and 13 (0.3%) women had this condition (*p* = 0.325). A total of 18 (0.6%) men and 10 (0.2%) women had disseminated intravascular coagulation (*p* = 0.012).

### 3.3. Subgroup Analyses According to Age, Diabetes Mellitus, or Hypertension

The number of men and women aged <35 years, 35–49 years, 50–64 years, and ≥65 years was 1133 and 1282, 486 and 916, 793 and 1354, and 552 and 811, respectively. In patients aged <35 years, mortality was not observed among both sexes. In patients aged 35–49 years, the 40-days survival rates were 99.2% and 100% for men and women, respectively (*p* = 0.284). In patients aged 50–64 or ≥65 years, the survival rates were greater for women than for men (*p* < 0.001; Appendix A). Multivariate Cox regression analyses showed favorable results for women in the subgroup analysis according to age (50–64 or ≥65 years), DM, or hypertension (Appendix A).

Appendix A shows the clinical outcomes according to the age groups. We noted favorable results for the use of inotropes, conventional oxygen therapy, HFNC, MV, and cardiac arrest for women in the 50–64 or ≥65-years age groups. Multivariate logistic regression analyses showed favorable results for women with respect to most complications (except AKI, ECMO, and AHF in the two age groups and MI in the 50–64-years age group) in the two age groups (Appendix A). Subgroup analysis for DM showed that most complications (except AKI, ECMO, and AHF in groups with and without DM and MI in the group without DM) were associated with favorable results for women. Subgroup analysis for hypertension showed that most complications (except ECMO, MI, and AHF in groups with and without hypertension and AKI in the group with hypertension) were associated with favorable results for women.

### 3.4. Comorbidities According to Sex and Age

Appendix A shows the rate of DM or hypertension according to sex and age. The rate of DM in the population aged 50–64 years was higher among men than among women. The prevalence of hypertension in the populations aged <35, 35–49, or 50–64 years was higher among men than among women. The CCI score was 0.4 ± 0.8 and 0.5 ± 0.8, 0.8 ± 1.4 and 0.9 ± 1.4, 1.6 ± 1.9 and 1.6 ± 1.8, and 3.0 ± 2.5 and 2.9 ± 2.4 in men and women aged <35, 35–46, 50–64, and ≥65 years, respectively. The CCI score increased with age, but there were no significant differences in the CCI score between men and women in the same age group.

## 4. Discussion

Our results can be helpful to expand the understanding of the effect of sex on the clinical outcomes of COVID-19 infection (especially the elderly population, regardless of the effect of estrogen). A previous study using a representative sample in South Korea showed that the mean age of menopause was 49 years [13]. Mortality in people aged <35 years or 35–49 years, in whom premenopausal patients were included, cannot be evaluated due to the low mortality in COVID-19 patients. Some women aged 50–64 years or almost all women aged ≥65 years can be considered menopausal. In 2019, before COVID-19, the mortality rate of the general population was 0.56% in men aged 50–64 years, 0.20% in women aged 50–64 years, 3.33% in men aged ≥65 years, and 2.58% in women aged ≥65 years [14]. The mortality rate in patients with COVID-19 infection was 2.65% in men aged 50–64 years, 0.37% in women aged 50–64 years, 17.57% in men aged ≥65 years, and 11.96% in women aged ≥65 years. For patients aged 50–64 years, the excess death rate was 2.09% in men and 0.17% in women. For patients aged ≥65 years, it was 14.24% in men and 9.38% in women. Therefore, the excess death rate in women aged 50–64 years was 8.1% that of men from the same age group and, in women aged ≥65 years, it was 65.9% that of men from the same age group. For the general population without COVID-19 infection, the odds ratio of death in men was 1.298 in those aged 50–64 years and 1.299 in those aged ≥65 years compared to women from the same age groups. For COVID-19 patients, the odds ratio of death in men was 7.339 in those aged 50–64 years and 1.569 in those aged ≥65 years compared to women from same age groups. Although we did not compare the mortality rate in participants without COVID-19 infection in the same period as the COVID-19 patients, COVID-19 infection may lead to higher mortality in men than in women, who may be postmenopausal. However, these beneficial effects of female sex were decreased in the elderly population, which was consistent with the results from Cagnacci’s study [15]. A data set with a longer follow-up, a greater number of deaths in a younger population, and a smaller age interval may reveal a greater difference in mortality between sexes in younger population groups.

Previous studies have reported that male patients with COVID-19 had poorer outcomes compared to female patients [3,4,5,6]. Yanez et al. analyzed data for confirmed cases and death from 16 countries and showed the mortality rate increased drastically after ≥65 years of age; moreover, higher mortality rates were seen in men than in women in all 16 countries included in the study: Austria, Belgium, Brazil, Canada, China, France, Germany, Italy, South Korea, Netherlands, Portugal, Spain, Sweden, Switzerland, United Kingdom, and the United States of America [16]. They showed that men infected with COVID-19 had a 1.77-fold higher mortality than women. However, the study presented descriptive results alone. Adjustments for comorbidities or subgroup analyses were not performed. A recent meta-analysis using 58 studies showed that men had a 1.57-fold higher odds ratio for mortality and a 1.65-fold higher odds ratio for severe infection than women [17]. Cagnacci used the data from the Italian National Institute of Health and compared the mortality in COVID-19 affected patients with that in patients unaffected by COVID-19 in the same period [15]. In analyses according to age groups within the same sex, a significantly higher death rate from COVID-19 was predominantly observed at ≥50 years old in women and ≥30 years old in men. They showed that men in most age groups had a higher mortality rate than women. However, favorable outcomes in women compared to men were predominantly greater in those aged 20–59 years than in those aged 60–89 years. These results reveal that pre-menopausal women would have the best outcome on analyses according to sex and age groups, which may lead to determining the importance of the effect of estrogen when studying the association between sex and mortality in COVID-19 infection.

The favorable outcomes in women may be attributed to various factors. First, high expressions of ACE-2 and transmembrane serine protease-2 (TMPRSS2) in men compared to women would be associated with poorer outcomes with respect to COVID-19. ACE-2 is expressed in various tissues, including the cardiovascular system, kidney, or lung. SARS-CoV-2 enters the cell through ACE-2 receptor. Liu et al. investigated an experimental study using gonadectomized mice with or without estrogen therapy and showed that the expression of renal ACE-2 was decreased by estrogen therapy [18]. Stelzig et al. showed that estrogen was associated with low expression of ACE-2 in human bronchial epithelial cells [19]. The increase in ACE-2 expression may be associated with the development of severe COVID-19 infection; however, opposing views exist. The up-regulation of angiotensin II, produced by ACE, leads to vasoconstriction, pro-fibrosis, and pro-inflammation in various tissues [20]. ACE-2 is associated with favorable effects via the down-regulation of the renin-angiotensin system and deactivation of angiotensin II. Once the virus enters through ACE-2, the virus induces a decrease in ACE-2 expression in the organ, which further induces more severe inflammatory injuries [21]. Bukowska et al. investigated the pacing model using human atrial tissues and showed estrogen lowered the ratio of ACE/ACE-2 from 2.12 ± 0.27 to 0.92 ± 0.21 at the transcriptional level and from 2.14 ± 0.3 to 0.94 ± 0.20 at the protein level [22]. The increase in ACE-2 expression by estrogen may be associated with attenuation of pro-inflammatory responses after COVID-19 infection. The association between ACE-2 expression and severity of COVID-19 infection is complex according to the disease course, and the key question remains unanswered. TMPRSS2 is a critical protease associated with viral spread through cleavage of the spike protein of SARS-CoV-2 [23]. Expression of TMPRSS2 would be associated with the activity of the androgen receptor, which may lead to high expression of TMPRSS2 in men. Therefore, the difference in ACE-2 and TMPRSS2 expression between the sexes may be associated with the severity or prognosis of COVID-19 patients. However, there are few studies regarding the differences in the two enzymes by sex and the impact of clinical outcomes.

Second, sex differences in innate or humoral immunities after viral infection may be associated with severity in COVID-19. Although there were few clinical data for the sex differences in immunity after COVID-19, previous data for a response to other viral infections, vaccines, or autoimmunity suggested that immune responses to COVID-19 may be different between sexes. Women generally present with higher antibody production in response to viral infection, which is associated with the estrogen effect or inherent difference in B cell response after infection [7,24]. In addition, there was evidence for sex differences in cytokine production or gene expression in innate cell subsets [7]. Previous studies showed that women are associated with increased stimulation of toll-like receptor-7 and interferon production after viral infection compared to men [25,26]. These differences may lead to different responses for the removal of viruses, which result in sex differences with respect to disease severity or mortality in COVID-19 patients.

Third, comorbidities caused by sex or sex-related factors may influence the difference. To date, there are limited data regarding the different comorbidities according to sex, using disaggregated data of male or female COVID-19 patients. We compared the proportion of patients with DM or hypertension according to sex and age groups. Our study showed a higher prevalence of DM and hypertension among men in the 50–64 age group and in the <35, 35–49, or 50–64 age groups, respectively, than among women; however, the ≥65 age group with the highest mortality showed no differences with respect to DM and hypertension. Additionally, the CCI score as the merged indicator was not different between the two sexes. Both analyses using aggregated data and subgroup analyses for DM, hypertension, or age groups showed mostly favorable results in women. These findings reveal that comorbidities are associated with clinical outcomes in COVID-19 patients in both sexes, but the difference of underlying comorbidities may not lead to the sex differences in clinical outcomes in COVID-19 patients.

In our study, clinical outcomes were poor in men, but the incidence rate was higher in women than in men. According to the Korean Statistical Information Service in May 2020, the number of men and women in South Korea was 25,856,030 and 25,985,341, respectively. The incidence rate in men and women was approximately 0.0115% and 0.0168%, respectively [27]. The World Health Organization merged each disaggregated data and reported that the distribution of infection between women and men is relatively similar [28]. The Global Health 50/50 research initiatives also presented that 55 among 92 countries identifiable with incidence by sex showed male predominance [6]. In our study, it is not clear whether female dominance in COVID-19 infection is associated with biological effect or gender-related factors. Considering male predominance from biological data, gender related factors such as life-style, socioeconomic status, or religious gathering may have a greater influence on female predominance in South Korea. Our study enrolled the participants using 15 May 2020 as the index date. In South Korea, the first outbreak was associated with a religious place [29]. Most of the cases in our study were patients from the first outbreak. Members of this religion were predominantly female. Therefore, in our study, female predominance was associated with this specific outbreak. After the first outbreak, sporadic cases were sustained. On 6 December 2020, the incidence of COVID-19 was similar between sexes (18,007 (48.0%) in men and 19,537 (52.0%) in women) [30]. Besides biologic factors, gender-related factors are also crucial to incidence or prognosis of COVID-19 [31]. The alcohol intake or number of cigarettes smoked was generally greater in men than in women, which may be another factor leading to gender differences in the clinical outcomes [32].

Our study has several limitations. First, our study was performed using health insurance claims with procedural or diagnostic codes from HIRA. Therefore, our study did not include laboratory or clinical data. The dataset has the possibility of over or under coding, which may lead to discrepancies between the relevant code and actual disease. The number of confirmed cases does not exactly coincide with the number of people using the HIRA system. For example, cumulative confirmed cases were exactly 9868 on 15 May 2020 (the index date), but the total confirmed cases using the HIRA system was 7590, at the same time. Beyond simple numbers of patients, if claims are not submitted in time to the HIRA system, the data for medical care may differ with the actual medical care. The differences in numbers of confirmed cases or patients’ prognosis can be influenced by the interval between timing of prescription and timing of claims. These discrepancies between data from claims and actual diagnosis or medical care can lead to selection bias. Population-based studies using claim data has the merit of providing researchers with a large sample size, but the abovementioned limitations are also present. Second, our study did not identify the causal relationship by sex. Our study analyzed insurance data alone and did not include data for biologic effects such as ACE2, TMPRSS2 levels, or data for immunologic status. Besides, gender-related factors were not included. Therefore, we demonstrated that men had poor clinical outcomes compared to women, especially the elderly. However, we could not identify the cause of sex differences in COVID-19. Third, we did not demonstrate sex differences in the younger population. In our study, the populations aged <35 years or 35–49 years had better outcomes compared to those aged 50–64 years or ≥65 years. We did not perform analyses beyond simple descriptive analysis results for the young population. Fourth, our study used claims data alone and did not include the data for laboratory findings, vital signs, or physical examinations at the admission or during follow-up. Fifth, our data did not include cause of death. However, in our study, most deaths were caused by infection-related complications, such as sepsis or acute respiratory distress syndrome, and the proportions of cause of death may not differ between sexes. These should be obvious confounding factors for predicting the prognosis in COVID-19 patients.

In conclusion, our population-based study showed that female patients with COVID-19 were associated with favorable outcomes, including survival. The impact of sex was more evident in the population aged 50–64 or ≥65 years. Further studies, including laboratory investigations or evaluation of sex-related factors, are required to identify the causal relationship and impact of sex in the young population.

## Figures and Tables

**Figure 1 jcm-10-00038-f001:**
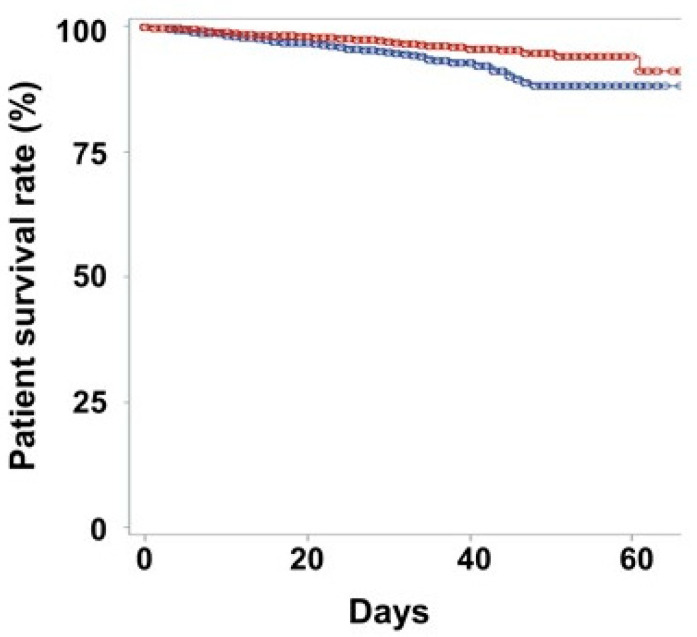
Kaplan–Meier survival curves, according to sex. The blue line shows the survival graph for men, and the red line shows the graph for women. The circle reveals the censored point. The 20-day survival rates for men and women were 96.6% and 97.9%, respectively (*p* < 0.001). The 40-day survival rates for men and women were 92.8% and 95.5%, respectively.

**Table 1 jcm-10-00038-t001:** Baseline characteristics of the participants.

	Total (*n* = 7327)	Men (*n* = 2964)	Women (*n* = 4363)	*p*-Value
Age (years)	47.0 ± 19.0	45.7 ± 19.3	47.9 ± 18.7	0.093
Follow-up duration (days)	20.9 ± 13.1	21.0 ± 13.3	20.7 ± 13.0	0.254
Myocardial infarction	97 (1.32%)	58 (2.2%)	39 (0.9%)	<0.001
Congestive heart failure	292 (3.99%)	128 (4.3%)	164 (3.8%)	0.229
Peripheral vascular disease	570 (7.78%)	206 (7.0%)	364 (8.3%)	0.029
Cerebrovascular disease	518 (7.07%)	218 (7.4%)	300 (6.9%)	0.433
Dementia	443 (6.05%)	151 (5.1%)	292 (6.7%)	0.005
Chronic pulmonary disease	1670 (22.79%)	609 (20.5%)	1061 (24.3%)	<0.001
Connective tissue disease	268 (3.66%)	80 (2.7%)	188 (4.1%)	<0.001
Peptic ulcer disease	1099 (15.00%)	387 (13.1%)	712 (16.3%)	<0.001
Mild liver disease	1540 (21.02%)	606 (20.4%)	934 (21.4%)	0.321
Diabetes mellitus	1223 (16.7%)	515 (17.4%)	708 (16.2%)	0.196
Hemiplegia	99 (1.35%)	50 (1.7%)	49 (1.1%)	0.040
Any malignancy	358 (4.89%)	126 (4.3%)	232 (5.3%)	0.038
Moderate to severe liver disease	9 (0.12%)	5 (0.2%)	4 (0.1%)	0.356
Metastatic tumor	33 (0.45%)	11 (0.4%)	22 (0.5%)	0.404
Acquired immune deficieny syndrome	5 (0.07%)	3 (0.1%)	2 (0.04%)	0.373
Hypertension	1559 (21.28%)	673 (22.7%)	886 (20.3%)	0.014

Data are expressed as number (percentage) for categorical variables and mean ± standard deviation for continuous variables. The *p*-values for continuous variables were tested with the Student’s *t*-test between men and women, and the Pearson χ^2^ test or Fisher exact test was used to analyze categorical variables.

**Table 2 jcm-10-00038-t002:** Cox regression analysis of mortality according to the variables.

	Univariate		Multivariate	
	HR (95% CI)	*p*-Value	HR (95% CI)	*p*-Value
Age (per increase 1 year)	1.11 (1.10−1.12)	<0.001	1.10 (1.09−1.11)	<0.001
Sex (ref: women)	1.69 (1.30−2.20)	<0.001	2.06 (1.58−2.69)	<0.001
CCI score (per increase 1 score)	1.35 (1.30−1.39)	<0.001	1.14 (1.09−1.20)	<0.001
Hypertension	8.20 (6.11−11.01)	<0.001	1.34 (0.97−1.86)	0.079

The data are expressed as HR (95% CI). Multivariate analysis was adjusted for age, sex, CCI score, and hypertension. Abbreviations: CCI, Charlson comorbidity index; HR, hazard ratio; CI, confidence interval.

**Table 3 jcm-10-00038-t003:** Clinical outcomes according to sex.

	Total	Men	Women	*p*-Value
Acute kidney injury	27 (0.4%)	17 (0.6%)	10 (0.2%)	0.017
Inotropics	182 (2.5%)	100 (3.4%)	82 (1.9%)	<0.001
Conventional oxygen therapy	901 (12.3%)	429 (14.5%)	472 (10.8%)	<0.001
High flow nasal cannula	178 (2.4%)	101 (3.8%)	77 (1.8%)	<0.001
Mechanical ventilation	123 (1.7%)	71 (2.4%)	52 (1.2%)	<0.001
Extracorporeal membrane oxygenation	20 (0.3%)	11 (0.4%)	9 (0.2%)	0.184
Cardiac arrest	42 (0.6%)	29 (1.0%)	13 (0.3%)	<0.001
Myocardial infarction	256 (3.5%)	122 (4.1%)	134 (3.1%)	0.017
Acute heart failure	380 (5.2%)	157 (5.3%)	223 (5.1%)	0.725

The data are expressed as number (percentage).

**Table 4 jcm-10-00038-t004:** Logistic regression analysis of the clinical outcomes according to sex.

	Univariate		Multivariate	
	OR (95% CI)	*p*-Value	OR (95% CI)	*p*-Value
**Inotropics**				
Age (per increase 1 year)	1.08 (1.07−1.09)	<0.001	1.06 (1.05−1.08)	<0.001
Sex (ref: women)	1.82 (1.36−2.46)	<0.001	2.09 (1.53−2.85)	<0.001
CCI score (per increase 1 score)	1.42 (1.35−1.49)	<0.001	1.17 (1.10−1.24)	<0.001
Hypertension	7.68 (5.64−10.54)	<0.001	1.66 (1.15−2.42)	0.007
**Conventional oxygen therapy**				
Age (per increase 1 year)	1.08 (1.07−1.08)	<0.001	1.07 (1.06−1.07)	<0.001
Sex (ref: women)	1.40 (1.21−1.61)	<0.001	1.67 (1.43−1.96)	<0.001
CCI score (per increase 1 score)	1.41 (1.36−1.45)	<0.001	1.11 (1.07−1.15)	<0.001
Hypertension	5.44 (4.70−6.29)	<0.001	1.31 (1.09−1.57)	0.004
**HFNC**				
Age (per increase 1 year)	1.08 (1.07−1.10)	<0.001	1.07 (1.06−1.09)	<0.001
Sex (ref: women)	1.96 (1.46−2.66)	<0.001	2.32 (1.70−3.19)	<0.001
CCI score (per increase 1 score)	1.38 (1.32−1.45)	<0.001	1.12 (1.05−1.19)	<0.001
Hypertension	7.40 (5.43−10.18)	<0.001	1.55 (1.08−2.26)	0.020
**Mechanical ventilation**				
Age (per increase 1 year)	1.07 (1.06−1.08)	<0.001	1.06 (1.04−1.07)	<0.001
Sex (ref: women)	2.04 (1.44−2.93)	<0.001	2.28 (1.57−3.31)	<0.001
CCI score (per increase 1 score)	1.34 (1.27−1.42)	<0.001	1.10 (1.01−1.18)	0.020
Hypertension	7.47 (5.16−10.98)	<0.001	2.01 (1.29−3.17)	0.002
**ECMO**				
Age (per increase 1 year)	1.05 (1.03−1.08)	<0.001	1.03 (0.98−1.06)	0.083
Sex (ref: women)	1.80 (0.75−4.48)	0.191	1.84 (0.76−4.59)	0.179
CCI score (per increase 1 score)	1.30 (1.13−1.47)	<0.001	1.11 (0.91−1.31)	0.280
Hypertension	6.92 (2.83−18.44)	<0.001	2.89 (0.95−9.50)	0.068
**Acute kidney injury**				
Age (per increase 1 year)	1.06 (1.04−1.09)	<0.001	1.03 (1.01−1.06)	0.023
Sex (ref: women)	2.51 (1.17−5.70)	0.021	2.58 (1.19−5.89)	0.019
CCI score (per increase 1 score)	1.39 (1.25−1.53)	<0.001	1.21 (1.05−1.37)	0.006
Hypertension	8.88 (4.02−21.57)	<0.001	2.65 (1.01−7.58)	0.057
**Cardiac arrest**				
Age (per increase 1 year)	1.08 (1.06−1.10)	<0.001	1.06 (1.04−1.09)	<0.001
Sex (ref: women)	3.31 (1.75−6.59)	<0.001	3.66 (1.92−7.35)	<0.001
CCI score (per increase 1 score)	1.43 (1.31−1.55)	<0.001	1.24 (1.11−1.38)	<0.001
Hypertension	6.58 (3.64−13.05)	<0.001	1.21 (0.58−2.61)	0.613
**Myocardial infarction**				
Age (per increase 1 year)	1.01 (1.00−1.02)	0.013	1.00 (0.99−1.01)	0.535
Sex (ref: women)	1.36 (1.06−1.74)	0.017	1.37 (1.07−1.76)	0.014
CCI score (per increase 1 score)	1.13 (1.07−1.19)	<0.001	1.11 (1.05−1.19)	<0.001
Hypertension	1.38 (1.04−1.82)	0.025	0.98 (0.69−1.39)	0.906
**Acute heart failure**				
Age (per increase 1 year)	1.04 (1.03−1.04)	<0.001	1.03 (1.03−1.04)	<0.001
Sex (ref: women)	1.04 (0.84−1.28)	0.724	1.10 (0.89−1.36)	0.397
CCI score (per increase 1 score)	1.21 (1.16−1.26)	<0.001	1.05 (0.99−1.10)	0.082
Hypertension	2.82 (2.28−3.48)	<0.001	1.27 (0.98−1.65)	0.075

The data are expressed as OR (95% CI). Multivariate analysis was adjusted for age, sex, CCI score, and hypertension. Abbreviations: OR, odds ratio; CI, confidence interval; HFNC, high flow nasal cannula; ECMO, extracorporeal membrane oxygenation; CCI, Charlson comorbidity index.

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
