# Peer review of "Effect of Sex on Clinical Outcomes in Patients with Coronavirus Disease: A Population-Based Study"

_jcm, 2020, doi:10.3390/jcm10010038_

Round 1
Reviewer 1 Report
This a retrospective chart review of patients diagnosed with COVID-19, aiming at evaluating the association between sex and clinical outcome in a Korean population-based dataset.
The authors reported higher survival rate amongst younger patients < 50 years of age, with no difference between sex, with higher mortality in those >50 and with pre-existing comorbidities, among men.
The authors did not differentiate between patients hospitalized or those treated as outpatient. They also did not address other complications associated with COVID-19 (stroke, pulmonary embolism, DIC, etcc..), and the cause of death.
They reported higher "incidence" of hypertension, hemiplegia, or MI amongst men. Did the author mean higher prevalence of these conditions at baseline pre-existing conditions, or did men have higher incidence of "hemiplegia and "acute MI?
Also, what did they mean by hemiplegia? what is the condition that lead to hemiplegia? ischemic stroke, intracranial hemorrhage, COVID-encephalitis?
It is important to clarify, since risk of pulmonary embolism, MI and stroke is high due to coagulopathy and has been associated with worse outcome
Author Response
This a retrospective chart review of patients diagnosed with COVID-19, aiming at evaluating the association between sex and clinical outcome in a Korean population-based dataset. The authors reported higher survival rate amongst younger patients < 50 years of age, with no difference between sex, with higher mortality in those >50 and with pre-existing comorbidities, among men.
The authors did not differentiate between patients hospitalized or those treated as outpatient. They also did not address other complications associated with COVID-19 (stroke, pulmonary embolism, DIC, etcc..), and the cause of death.
Answer: Thank you for your comments. We have evaluated hospitalization or newly developed cerebrovascular disease, pulmonary embolism, or disseminated intravascular coagulation after COVID-19 diagnosis. The diagnosis of pulmonary embolism and disseminated intravascular coagulation after COVID-19 infection was evaluated using I26 and D65, respectively. The number of patients with hospitalization was 2,800 (94.5%) in men and 4,111 (94.2%) in women, respectively (P = 0.681). Therefore, we did not perform subgroup analysis based on hospitalization. Among patients without prevalent cerebrovascular disease, 15 (0.5%) men and 5 (0.1%) women had newly developed cerebrovascular disease (P = 0.002). The incidence of pulmonary embolism was 13 (0.4%) in men and 13 (0.3%) in women (P = 0.325). The number of patients with disseminated intravascular coagulation was 18 (0.6%) in men and 10 (0.2%) in women (P = 0.012). Unfortunately, our data did not include the cause of death. However, in our study, most deaths were caused by infection-related complications, such as sepsis or acute respiratory distress syndrome, and the proportion of cause of death may not differ between sexes. We have added these comments in the Methods, Results, and Limitations sections.
They reported higher "incidence" of hypertension, hemiplegia, or MI amongst men. Did the author mean higher prevalence of these conditions at baseline pre-existing conditions, or did men have higher incidence of "hemiplegia and "acute MI?
Answer: Thank you for your comments. We have revised “incidence” to “prevalence”.
Also, what did they mean by hemiplegia? what is the condition that lead to hemiplegia? ischemic stroke, intracranial hemorrhage, COVID-encephalitis?
Answer: Thank you for your comments. In our study, hemiplegia was considered as an underlying comorbidity, as we previously revised. The condition is a well-known comorbidity or disability, which is associated with high mortality regardless of cause of hemiplegia in many newly developed diseases. In addition, we have evaluated the cause of prevalent hemiplegia. We found that 11 (22%) men and 10 (20.4%) women developed hemiplegia due to hemorrhagic stroke. Moreover, 26 (52%) men and 33 (67.3%) women developed hemiplegia due to ischemic stroke. There was no significant difference in type of stroke in patients with prevalent hemiplegia between sexes (P = 0.178). We have added these comments in the Results section.
It is important to clarify, since risk of pulmonary embolism, MI and stroke is high due to coagulopathy and has been associated with worse outcome
Answer: Thank you for your comments. As the reviewer suggested, we have added the data regarding the incidence of pulmonary embolism, stroke, and DIC after COVID-19 infection. The detailed explanation is provided in the Answer to question 1.
Reviewer 2 Report
This article evaluates the risk of developing complications and death in COVID-19 Korean infected subjects divided by sex and of different age.
The study has merit but there are several points that need to be further considered:
Overall the study is confused and difficult to follow. I believe that the authors mislabeled the tables. For example Table 2 does not refer to survival but to mortality otherwise it seems the men die less than women.
The same applies to table 4 reporting OR. For any complication there is a OR with men as reference. All the odds ratio are above 1, as to indicate that female have an increased risk of each different procedure. It seems that the authors in their writing sustain the opposite.
In the abstract there what is written on line 20 is opposite to the conclusions.
It is not know why there were so many women in the study almost double the number of men, even if all the studies indicate that infection is similar in the two sexes.
As previously reported Am J Obstet Gynecol 2020; 223:453, the correct way to show mortality by COVID-19 is to correct mortality of infected individuals by mortality of non infected individuals on the basis of sex and age. In addition it should be useful to express mortality of women on that of men of different ages, because steroid hormones of women disappear after the menopause, and it would be useful to see whether the proportion of women death vs. that of men changes with hormonal status, as previously reported.
As for the effect of estrogens of ACE2, there are also data sustaining that estrogens stimulates ACE2. Exp Biol Med 2017; 242:1412
Author Response
Reviewer 2
This article evaluates the risk of developing complications and death in COVID-19 Korean infected subjects divided by sex and of different age. The study has merit but there are several points that need to be further considered:
Overall the study is confused and difficult to follow. I believe that the authors mislabeled the tables. For example Table 2 does not refer to survival but to mortality otherwise it seems the men die less than women.
The same applies to table 4 reporting OR. For any complication there is a OR with men as reference. All the odds ratio are above 1, as to indicate that female have an increased risk of each different procedure. It seems that the authors in their writing sustain the opposite.
In the abstract there what is written on line 20 is opposite to the conclusions.
Answer: Thank you for your comments. We apologize for the errors in our manuscript. We have revised some expressions, such as “incidence, survival, or men” as reviewer pointed. We have revised the abstract, manuscript, and Tables 2 and 4.
It is not know why there were so many women in the study almost double the number of men, even if all the studies indicate that infection is similar in the two sexes.
Answer: Thank you for your comments. Our study enrolled the participants using May 15, 2020 as the index date. In South Korea, the first outbreak was associated with a religious place [1]. Most of the cases in our study were patients from the first outbreak. Members of the religion were predominantly female. Therefore, in our study, female predominance was associated with a specific outbreak. After the first outbreak, sporadic cases were sustained. On December 6, 2020, the incidence of COVID-19 was similar between sexes (18,007 [48.0%] in men and 19,537 [52.0%] in women) [2]. We have added these comments in the Discussion section.
Added references
[1] Kang J, et al. South Korea's responses to stop the COVID-19 pandemic. Am J Infect Control. 48, 1080-1086 (2020).
[2] Korea Centers for Disease Control and Prevention. Cases in Korea. Available at: http://ncov.mohw.go.kr/bdBoardList_Real.do?brdId=1&brdGubun=11&ncvContSeq=&contSeq=&board_id=&gubun=. (2020).
As previously reported Am J Obstet Gynecol 2020; 223:453, the correct way to show mortality by COVID-19 is to correct mortality of infected individuals by mortality of non infected individuals on the basis of sex and age. In addition it should be useful to express mortality of women on that of men of different ages, because steroid hormones of women disappear after the menopause, and it would be useful to see whether the proportion of women death vs. that of men changes with hormonal status, as previously reported.
Answer: Thank you for your comments. A previous study using a representative sample in South Korea showed that the mean age of menopause was 49 years [1]. Mortality in people aged <35 years or 35-49 years, in whom premenopausal patients were included, cannot be evaluated due to the low mortality in COVID-19 patients. Almost all women aged 50-64 years or ≥65 years can be considered menopausal. In 2019, before COVID-19, the mortality rate of the general population was 0.56% in men aged 50-64 aged, 0.20% in women aged 50-64 years, 3.33% in men aged ≥65 years, and 2.58% in women aged ≥65 years [2]. The mortality rate in patients with COVID-19 infection was 2.6% in men aged 50-64 years, 0.4% in women aged 50-64 years, 17.6% in men aged ≥65 years, and 12.0% in women aged ≥65 years. For the general population without COVID-19 infection, the odds ratio of death in men was 1.298 in those aged 50-64 years and 1.299 in those aged ≥65 years compared to women from the same age groups. For COVID-19 patients, the odds ratio of death in men was 7.339 in those aged 50-64 years and 1.569 in those aged ≥ 65 years compared to women from same age groups. Although we did not compare the mortality rate in participants without COVID-19 infection in the same period as the COVID-19 patients, COVID-19 infection may lead to higher mortality in men than in women, who may be almost postmenopausal. We have added these comments in the Discussion section.
Added references
[1] Lee SS, Han KD, Joo YH. Association of perceived tinnitus with duration of hormone replacement therapy in Korean postmenopausal women: a cross-sectional study. BMJ Open. 2017 Jul 10;7(7):e013736.
[2] Korean Statistical Information Service. Major Indicators of Korea. Avaiable at: https://kosis.kr/statisticsList/statisticsListIndex.do?menuId=M_01_01&vwcd=MT_ZTITLE&parmTabId=M_01_01. (2020).
As for the effect of estrogens of ACE2, there are also data sustaining that estrogens stimulates ACE2. Exp Biol Med 2017; 242:1412
Answer: Thank you for your comments. We have reviewed the reference per the reviewer comments. Bukowska et al. showed that the administration of estrogen was associated with the down-regulation of ACE2/ACE gene expression in human atrial tissue [1]. We have added these comments in the Discussion section.
Added references
[1] Bukowska A, et al. Protective regulation of ACE2/ACE gene expression by estrogen in human atrial tissue from elderly men. Exp Biol Med (Maywood) 242, 1412-1423 (2017).

Reviewer 3 Report
This is a cross-sectional study aiming to investigate the association between sex and clinical outcomes in patients with COVID-19 using claims data from the Korea database. Clinical outcomes except from mortality was assessed using ICD10 codes.
Overall, I think the content and originality of the paper is low. The conclusion that female patients with COVID-19 have favorable outcomes and that the impact of sex is most evident in patients aged 50-64 and >=65 + years is not new. Thus, I have difficulties seeing what this study adds to the literature, which should be explained more explicit in the paper. In addition, I have the following comments:
What is the definition of clinical outcomes? For instance, stated in the first sentence in the abstract. It should be elaborated earlier in the manuscript.
Page 1, line 31: The number of deaths among the confirmed cases was approximately 285. I guess, you know exactly how many people died?
The study population is based on insurance claims data from the Health Insurance Review and Assessment Service of Korea. There is a short description about HIRA, but it is still difficult to find out what it represents when you do not know anything about HIRA. This information at page 2 could be elaborated.
I am wondering why 59,5% of the study population is women. Is there a sex difference in the amount of men and women who have insurance claims? This could potentially result in a selection bias of the study. Who is having this insurance claims? People with the worse health? If more women than men included in your study have bad health, your results based on sex differences will be biased towards null. Thus, you could expect even larger advantages for women than found in this study. Thus, I am not sure that this sample is representative of the general population of Korea. This should be elaborated further under limitations and which bias we would expect based on the insurance claims data.
And please give percentage in parentheses – a total of 7327 patients were included of these 2964 (xx%) were men and 4363 (xx % were women).
Then you abbreviate all your variables at page 2, however, the reader cannot remember all these abbreviations later in the manuscript, so I suggest that you write them out everywhere.
You do not describe the Carlson comorbidity index (CCI). The CCI index does not contain all important comorbidities. At least it should be described in detail how this index was constructed. For instance, you adjust for CCI score and hypertension – I guess it is because hypertension is important but not included in the CCI score.
Please explain at page 3 why you adjust for the chose covariates. It would have been interesting to see the results only adjusted for age.
You replicate a lot of findings that are already present in the literature e.g. sex differences in COVID-19 mortality. Here you should at least provide references to important studies showing differences in COVID-19 mortality between men and women in different age groups – also studies from other parts of the world for instance Europe.
In the discussion, you have a longer section about why women have better outcomes than men after/during COVID-19. However, these results seem to be based on the literature and not on your findings. It would be more interesting to start the discussion with what you find and relate it the what other researchers found.
Language: needs attention. There are several spelling mistakes and formulations that would benefit from being improved.
Author Response
Reviewer 3
This is a cross-sectional study aiming to investigate the association between sex and clinical outcomes in patients with COVID-19 using claims data from the Korea database. Clinical outcomes except from mortality was assessed using ICD10 codes. Overall, I think the content and originality of the paper is low. The conclusion that female patients with COVID-19 have favorable outcomes and that the impact of sex is most evident in patients aged 50-64 and >=65 + years is not new. Thus, I have difficulties seeing what this study adds to the literature, which should be explained more explicit in the paper. In addition, I have the following comments:
What is the definition of clinical outcomes? For instance, stated in the first sentence in the abstract. It should be elaborated earlier in the manuscript.
Answer: Thank you for your comments. We have added that clinical outcomes included hospitalization, the use of inotropics, high flow nasal cannula, conventional oxygen therapy, high flow nasal cannula, mechanical ventilation, extracorporeal membrane oxygenation, development of acute kidney injury, cardiac arrest, myocardial infarction, acute heart failure, pulmonary embolism, or disseminated intravascular coagulation after the diagnosis of COVID-19 in the Abstract and Method sections.
Page 1, line 31: The number of deaths among the confirmed cases was approximately 285. I guess, you know exactly how many people died?
Answer: Thank you for your comments. The exact number of deaths was 285. We have deleted “approximately”.
The study population is based on insurance claims data from the Health Insurance Review and Assessment Service of Korea. There is a short description about HIRA, but it is still difficult to find out what it represents when you do not know anything about HIRA. This information at page 2 could be elaborated.
Answer: Thank you for your comments. In South Korea, 97% of the population is obliged to enroll in the National Health Insurance program and pay the insurance according to income, regardless of medical care [1]. Patients who receive medical care, except non-essential care such as cosmetic surgery, pay approximately 5~30% of the total costs to the hospital that performs the relevant procedures. The hospital then submits the claim data to the Health Insurance Review & Assessment (HIRA) service, who reimburse the remaining costs (approximately 70~95% of the total cost) except the patient’s payments. The claim data includes diagnosis using ICD-10 code, procedures, prescription records, and simple demographic data. Most of the population not included in the the National Health Insurance program is included in the Medical Aid program. Patients included in the Medical Aid program do not pay the hospital; the hospital reimburses all costs through claims to HIRA using the same approach as that used by the National Health Insurance program. Claim data were not originally developed for medical studies. However, all hospitals and almost the entire population in South Korea uses the HIRA system. The data in the HIRA system includes numerous demographics and diagnostic codes. If claims are not submitted to the HIRA system in time, the data for medical care do not coincide with actual medical care. However, many studies have used the data for conducting population-based study. We have added these comments in the Methods section.
[1] Kim HK, et al. Data Configuration and Publication Trends for the Korean National Health Insurance and Health Insurance Review & Assessment Database. Diabetes Metab J. 44, 671-678 (2020).
I am wondering why 59,5% of the study population is women. Is there a sex difference in the amount of men and women who have insurance claims? This could potentially result in a selection bias of the study. Who is having this insurance claims? People with the worse health? If more women than men included in your study have bad health, your results based on sex differences will be biased towards null. Thus, you could expect even larger advantages for women than found in this study. Thus, I am not sure that this sample is representative of the general population of Korea. This should be elaborated further under limitations and which bias we would expect based on the insurance claims data.
Answer: Thank you for your comments. As the reviewer pointed out, number of confirmed cases do not exactly coincide with the number of people using the HIRA system. For example, cumulative confirmed cases were exactly 9,868 on May 15, 2020 (the index date), but total confirmed cases using the HIRA system were 7,590, at same time. Beyond simple numbers of patients, if claims are not submitted in time to the HIRA system, the data for medical care may differ with actual medical care. The differences in numbers of confirmed cases or patients’ prognosis can be influenced by interval between timing of prescription and timing of claims. These discrepancies between data from claims and actual diagnosis or medical care can lead to selection bias. Population-based studies using claim data has the merit of providing researchers with a large sample size, but the abovementioned limitations are also present. We have added these comments in the Discussion section.
And please give percentage in parentheses – a total of 7327 patients were included of these 2964 (xx%) were men and 4363 (xx % were women).
Answer: Thank you for your comments. We have added the percentage of patients for each sex in the Abstract and Results sections.
Then you abbreviate all your variables at page 2, however, the reader cannot remember all these abbreviations later in the manuscript, so I suggest that you write them out everywhere.
Answer: Thank you for your comments. We have added Supplementary Table 1 for abbreviations in our study.
You do not describe the Carlson comorbidity index (CCI). The CCI index does not contain all important comorbidities. At least it should be described in detail how this index was constructed. For instance, you adjust for CCI score and hypertension – I guess it is because hypertension is important but not included in the CCI score.
Answer: Thank you for your comments. As reviewer pointed out, we have added detailed comments regarding the CCI score without hypertension in the Methods section.
Please explain at page 3 why you adjust for the chose covariates. It would have been interesting to see the results only adjusted for age.
Answer: Thank you for your comments. Clinical outcomes in acute or chronic diseases are highly influenced by underlying comorbidities, such as diabetes mellitus, hypertension, cerebrovascular disease, or heart disease. Although our study focused on the impact of sex in prognosis in COVID-19 patients, adjustment for underlying comorbidities will be important for identify whether sex is independently associated with prognosis. Therefore, we used CCI score as a merged indicator. Most comorbidities associated with prognosis were included in CCI score, but hypertension as an important disease was not included in the CCI score. Therefore, we consider hypertension as an additional covariate. We have added these comments in the Statistical analyses section.
You replicate a lot of findings that are already present in the literature e.g. sex differences in COVID-19 mortality. Here you should at least provide references to important studies showing differences in COVID-19 mortality between men and women in different age groups – also studies from other parts of the world for instance Europe.
Answer: Thank you for your comments. Yanez et al. analyzed data for confirmed cases and death from 16 countries and showed the mortality rate increased drastically after ≥65 years of age; higher mortality rates were seen in men than in women in all 16 countries included in the study: Austria, Belgium, Brazil, Canada, China, France, Germany, Italy, South Korea, Netherlands, Portugal, Spain, Sweden, Switzerland, United Kingdom, and the United States of America [1]. They showed that men infected with COVID-19 had 1.77-fold higher mortality than women. However, the study presented descriptive results alone. Adjustments for comorbidities or subgroup analyses were not performed. A recent meta-analysis using 58 studies showed that men had a 1.57-fold higher odds ratio for mortaliy and a 1.65-fold higher odds ratio for severe infection than women [2]. Cagnacci used the data from the Italian National Institute of Health and compared the mortality in COVID-19 affected patients with that in patients unaffected by COVID-19 in the same period [3]. On analyses accroding to age groups within the same sex, a significantly higher death rate from COVID-19 was predominantly observed at ≥50 years old in women and ≥30 years old in men. They showed that men in most age groups had a higher mortality rate than women. However, favorable outcomes in women compared to men were predominantly greater in those aged 20-59 years than in those aged 60-89 years. These results reveal that pre-menopausal women would have the best outcome on analyses according to sex and age groups, which may lead to determining the importance of the effect of estrogen when studying the association between sex and mortality in COVID-19 infection. We have added these comments in the Discussion section
Added references
[1] Yanez ND, et al. COVID-19 mortality risk for older men and women. BMC Public Health. 20, 1742 (2020).
[2] Izcovich A, et al. Prognostic factors for severity and mortality in patients infected with COVID-19: A systematic review. PLoS One. 15, e0241955 (2020).
[3] Cagnacci A, Xholli A. Age-related difference in the rate of coronavirus disease 2019 mortality in women versus men. Am J Obstet Gynecol. 223, 453-454 (2020).
In the discussion, you have a longer section about why women have better outcomes than men after/during COVID-19. However, these results seem to be based on the literature and not on your findings. It would be more interesting to start the discussion with what you find and relate it the what other researchers found.
Answer: Thank you for your comments. Our results can be helpful to expand the understanding for the effect of sex on clinical outcomes in COVID-19 infection (especially the elderly population, regardless of the effect of estrogen). A previous study using a representative sample in South Korea showed that the mean age of menopause was 49 years [1]. Mortality in people aged <35 years or 35-49 years, in whom premenopausal patients were included, cannot be evaluated due to the low mortality in COVID-19 patients. Almost all women aged 50-64 years or ≥65 years can be considered menopausal. In 2019, before COVID-19, the mortality rate of the general population was 0.56% in men aged 50-64 years, 0.20% in women aged 50-64 years, 3.33% in men aged ≥65 years, and 2.58% in women aged ≥65 years [2]. The mortality rate in patients with COVID-19 infection was 2.6% in men aged 50-64 years, 0.4% in women aged 50-64 years, 17.6% in men aged ≥65 years, and 12.0% in women aged ≥65 years. For the general population without COVID-19 infection, the odds ratio of death in men was 1.298 in those aged 50-64 years and 1.299 in those aged ≥65 years compared to women from the same age groups. For COVID-19 patients, the odds ratio of death in men was 7.339 in those aged 50-64 years and 1.569 in those aged ≥ 65 years compared to women from same age groups. Although we did not compare the mortality rate in participants without COVID-19 infection in the same period as the COVID-19 patients, COVID-19 infection may lead to higher mortality in men than in women, who may be almost postmenopausal. A data set with a longer follow-up and greater deaths in a younger population may reveal a greater difference in mortality between sexes in younger population groups. We have added these comments in the first paragraph of the Discussion section.
Added references
[1] Lee SS, Han KD, Joo YH. Association of perceived tinnitus with duration of hormone replacement therapy in Korean postmenopausal women: a cross-sectional study. BMJ Open. 2017 Jul 10;7(7):e013736.
[2] Korean Statistical Information Service. Major Indicators of Korea. Avaiable at: https://kosis.kr/statisticsList/statisticsListIndex.do?menuId=M_01_01&vwcd=MT_ZTITLE&parmTabId=M_01_01. (2020).
Language: needs attention. There are several spelling mistakes and formulations that would benefit from being improved.
Answer: Thank you for your comments. To address the reviewer’s concern, two native English-speaking scientific editors have rechecked the manuscript for language and revised such ambiguous sentences throughout the manuscript.

Round 2
Reviewer 2 Report
Manuscript greatly improved and almost ready for publication. Nevertheless the paper by Bukoska is misquoted. The paper does not report a dowregulation but an uperegulation of ACE2 in human atrial tissue. Herein I enclose the sentences of the original paper "...estrogen increased the amounts of angiotensin-converting enzyme 2-mRNA (1.890.23; P<0.05) but reduced that of angiotensin-converting enzyme-mRNA (0.780.07, P<0.05)." Thus estrogens stimulate and do not down regulate ACE2, as sustained by the authors of this manuscript. What estrogens down regulate is the ratio ACE/ACE2 as reported in the following sentence "...administration of estrogen substantially lowered the angiotensin-converting enzyme/angiotensin-converting enzyme 2 ratio at the transcript (0.920.21 vs. 2.120.27 at 4 Hz) and protein level (0.940.20 vs. 2.140.3 at 4 Hz)." This should be appropriately addressed in the manuscript and discussed in the proposed pathogenetic mechanisms. Please consider that ACE2 is important in inactivating angiotensin and its negative vascular and inflammatory effects.
The authors presented data on mortality in COVID and not COVID affected populations requested. I made same calculations on their data and obtained the following. In the group 50-64 yrs. excess death rate of infected women (on non infected) was 0.2% and of men 2.04%. In the group >65 yrs excess death rate of women was 9.42% and that of men 14.27%. Accordingly in the 50-64 years of age death rate of women was 9.8% that of men and in the group > 65 years age mortality of women was 66% that of men. This difference seems rather impressive. Even if mean age at menopause is 49, half of the gaussian curve lies in the years over 49 years of age. More strict age intervals would have helped to identify when mortality of women vs. men abruptly rises, but the data do not seem very different from those previously published.
Author Response
Manuscript greatly improved and almost ready for publication. Nevertheless the paper by Bukoska is misquoted. The paper does not report a dowregulation but an uperegulation of ACE2 in human atrial tissue. Herein I enclose the sentences of the original paper "...estrogen increased the amounts of angiotensin-converting enzyme 2-mRNA (1.890.23; P<0.05) but reduced that of angiotensin-converting enzyme-mRNA (0.780.07, P<0.05)." Thus estrogens stimulate and do not down regulate ACE2, as sustained by the authors of this manuscript. What estrogens down regulate is the ratio ACE/ACE2 as reported in the following sentence "...administration of estrogen substantially lowered the angiotensin-converting enzyme/angiotensin-converting enzyme 2 ratio at the transcript (0.920.21 vs. 2.120.27 at 4 Hz) and protein level (0.940.20 vs. 2.140.3 at 4 Hz)." This should be appropriately addressed in the manuscript and discussed in the proposed pathogenetic mechanisms. Please consider that ACE2 is important in inactivating angiotensin and its negative vascular and inflammatory effects.
Answer: Thank you for your comments. We have revised the relevant sentences as follows:
“The increase in ACE-2 expression may be associated with the development of severe COVID-19 infection; however, opposing views exist. The up-regulation of angiotensin II, produced by ACE, leads to vasoconstriction, pro-fibrosis, and pro-inflammation in various tissues [1]. ACE-2 is associated with favorable effects via the down-regulation of the renin-angiotensin system and deactivation of angiotensin II. Once the virus enters through ACE-2, the virus induces a decrease in ACE-2 expression in the organ, which further induces more severe inflammatory injuries [2]. Bukowska et al. investigated the pacing model using human atrial tissues and showed that estrogen lowered the ratio of ACE/ACE-2 from 2.12 ± 0.27 to 0.92 ± 0.21 at the transcriptional level and from 2.14 ± 0.3 to 0.94 ± 0.20 at the protein level [3]. The increase in ACE-2 expression by estrogen may be associated with attenuation of pro-inflammatory responses after COVID-19 infection. The association between ACE-2 expression and severity of COVID-19 infection is complex according to the disease course, and the key question remains unanswered.” We have added these statements to the Discussion section.
Added references
[1] Vaduganathan M, et al. Renin-Angiotensin-Aldosterone System Inhibitors in Patients with Covid-19. N Engl J Med. 382, 1653-1659 (2020).
[2] Kuba K, Imai Y, Penninger JM. Angiotensin-converting enzyme 2 in lung diseases. Curr Opin Pharmacol. 6, 271-276 (2006).
[3] Bukowska A, et al. Protective regulation of the ACE2/ACE gene expression by estrogen in human atrial tissue from elderly men. Exp Biol Med (Maywood). 242, 1412-1423 (2017).
The authors presented data on mortality in COVID and not COVID affected populations requested. I made same calculations on their data and obtained the following. In the group 50-64 yrs. excess death rate of infected women (on non infected) was 0.2% and of men 2.04%. In the group >65 yrs excess death rate of women was 9.42% and that of men 14.27%. Accordingly in the 50-64 years of age death rate of women was 9.8% that of men and in the group > 65 years age mortality of women was 66% that of men. This difference seems rather impressive. Even if mean age at menopause is 49, half of the gaussian curve lies in the years over 49 years of age. More strict age intervals would have helped to identify when mortality of women vs. men abruptly rises, but the data do not seem very different from those previously published.
Answer: Thank you for your comments. We have revised the first paragraph of the Discussion section accordingly. The revisions are highlighted in red font in the revised manuscript.
“Our results can be helpful to expand the understanding of the effect of sex on the clinical outcomes in COVID-19 infection (especially the elderly population, regardless of the effect of estrogen). A previous study using a representative sample in South Korea showed that the mean age of menopause was 49 years. Mortality in people aged <35 years or 35-49 years, in whom premenopausal patients were included, cannot be evaluated due to the low mortality in COVID-19 patients. Some women aged 50-64 years or almost all women aged ≥65 years can be considered menopausal. In 2019, before COVID-19, the mortality rate of the general population was 0.56% in men aged 50-64 years, 0.20% in women aged 50-64 years, 3.33% in men aged ≥65 years, and 2.58% in women aged ≥65 years [14]. The mortality rate in patients with COVID-19 infection was 2.65% in men aged 50-64 years, 0.37% in women aged 50-64 years, 17.57% in men aged ≥65 years, and 11.96% in women aged ≥65 years (data not shown). For patients aged 50-64 years, the excess death rate was 2.09% in men and 0.17% in women. For patients aged ≥65 years, it was 14.24% in men and 9.38% in women. Therefore, the excess death rate in women aged 50-64 years was 8.1% that of men from the same age group and, in women aged ≥65 years, it was 65.9% that of men from the same age group. For the general population without COVID-19 infection, the odds ratio of death in men was 1.298 in those aged 50-64 years and 1.299 in those aged ≥65 years compared to women from the same age groups. For COVID-19 patients, the odds ratio of death in men was 7.339 in those aged 50-64 years and 1.569 in those aged ≥ 65 years compared to women from same age groups. Although we did not compare the mortality rate in participants without COVID-19 infection in the same period as the COVID-19 patients, COVID-19 infection may lead to higher mortality in men than in women, who may be postmenopausal. However, these beneficial effects of female sex were decreased in the elderly population, which was consistent with the results from Cagnacci’s study. A data set with a longer follow-up, greater deaths in a younger population, and smaller age interval may reveal a greater difference in mortality between sexes in younger population groups.”

Reviewer 3 Report
The authors have addressed all my comments and concerns from the first version.
Author Response
The authors have addressed all my comments and concerns from the first version.
Answer: Thank you for your helpful comments. We believe that our results can be helpful in expand the understanding of the association between sex and prognosis of COVID-19 patients.
